# Modulation of GCN2/eIF2α/ATF4 Pathway in the Liver and Induction of FGF21 in Young Goats Fed a Protein- and/or Phosphorus-Reduced Diet

**DOI:** 10.3390/ijms24087153

**Published:** 2023-04-12

**Authors:** Sarah L. Weber, Karin Hustedt, Nadine Schnepel, Christian Visscher, Alexandra S. Muscher-Banse

**Affiliations:** 1Institute for Physiology and Cell Biology, University of Veterinary Medicine Hannover, 30173 Hannover, Germany; 2Institute for Animal Nutrition, University of Veterinary Medicine Hannover, 30173 Hannover, Germany

**Keywords:** amino acids, reduced-protein diet, reduced-phosphorus diet, ruminants, FGF21, GCN2, ATF4

## Abstract

Mammals respond to amino acid (AA) deficiency by initiating an AA response pathway (AAR) that involves the activation of general control nonderepressible 2 (GCN2), phosphorylation of eukaryotic translation initiation factor 2α (eIF2α), and activation of transcription factor 4 (ATF4). In this study, the effects of protein (N) and/or phosphorus (P) restriction on the GCN2/eIF2α/ATF4 pathway in the liver and the induction of fibroblast growth factor 21 (FGF21) in young goats were investigated. An N-reduced diet resulted in a decrease in circulating essential AA (EAA) and an increase in non-essential AA (NEAA), as well as an increase in hepatic mRNA expression of *GCN2* and *ATF4* and protein expression of GCN2. Dietary N restriction robustly increased both hepatic *FGF21* mRNA expression and circulating FGF21 levels. Accordingly, numerous significant correlations demonstrated the effects of the AA profile on the AAR pathway and confirmed an association. Furthermore, activation of the AAR pathway depended on the sufficient availability of P. When dietary P was restricted, the GCN2/eIF2α/ATF4 pathway was not initiated, and no increase in FGF21 was observed. These results illustrate how the AAR pathway responds to N- and/or P-reduced diets in ruminants, thus demonstrating the complexity of dietary component changes.

## 1. Introduction

From an ecological and economic perspective in the livestock industry, reducing the content of crude protein (CP) and phosphorus (P) in the diet is an effective approach to reduce nitrogen and P excretion by animals. In this way, harmful effects on the environment are avoided, and feed resources are saved [1,2,3]. Nevertheless, a sufficient quantity and quality of protein in the diet as well as an adequate amount of P are necessary to meet nutrient requirements. However, it should be noted that ruminants recycle endogenous inorganic phosphate (P_i_) to maintain P homeostasis [4,5]. Furthermore, the ruminohepatic urea cycle in ruminants is an effective mechanism for N recycling that can cope with reduced N availability without having any negative effects on performance as long as energy supply is ensured [6,7]. Considering that dietary proteins are key sources of essential amino acids (EAA), while non-essential amino acids (NEAA) can be synthesized endogenously, the pathway of protein utilization by the ruminants should be considered from a different perspective. Dietary protein is degraded by rumen microorganisms to ammonia, peptides, and free AA [7]. They are used by these microbes as a source of N for protein synthesis. Microbial crude protein, the protein content of ruminal microorganisms that passes through and is absorbed by the small intestine, is the most important source of protein and thus contributes significantly to the overall AA supply of ruminants [8]. Microbial performance in the rumen is closely related to intracellular and extracellular AA homeostasis, partly through microbial degradation capacity and partly through the ability of the microbial population itself to synthesize EAA and NEAA de novo. Therefore, determining protein or AA requirements in ruminants is a difficult task because of the rumen microflora present. However, a low-protein (LP) diet may lead to a reduction in certain cellular EAA regardless of the species.

In non-ruminants, the GCN2/eIF2α/ATF4 pathway is often described and is commonly referred to as the amino acid response (AAR) pathway. This pathway reduces global protein synthesis as part of the adaptive response to AA deprivation, allowing cells to conserve resources while adopting a new gene expression program to avoid stress damage [9]. In ruminants, the effects of LP diets on this metabolic pathway are still unknown. We therefore want to investigate whether there are comparable mechanisms in ruminants despite the specific microbial flora. In non-ruminants, AA deficiency, or more precisely AA imbalance, is known to be recognized by general control nonderepressible 2 (GCN2), a kinase that phosphorylates eukaryotic initiation factor 2 (eIF2) on its alpha subunit [10]. Moreover, eIF2α ~P enhances the translation of activating transcription factor 4 (ATF4), whose target genes are involved in the adaption to nutritional stress, such as the upregulation of AA biosynthesis and AA transporters [11,12], to restore cellular AA homeostasis.

The peptide hormone fibroblast growth factor 21 (FGF21) is induced by elevated ATF4 concentrations in the liver and is released into the bloodstream [13]. FGF21 is synthesized by various organs [14] but is secreted mainly by the liver [15] in response to nutritional and metabolic challenges [16]. It was originally thought to play a key role in mediating metabolic responses to fasting or starvation, including fatty acid oxidation and ketogenesis [17], but it can also be triggered by an inadequate supply of protein or AA in the diet, where it is required for adaptive metabolic changes [13,18]. Firmenich and colleagues demonstrated that hepatic expression of *FGF21* mRNA was significantly increased in young goats fed an LP diet [19], suggesting that dietary protein restriction also leads to induction of the AAR pathway in ruminants. 

In this study, the effects of N and P restriction on the AAR pathway and the induction of FGF21 in the livers of young goats were investigated. In addition, the interaction between the effects of N- and P-restricted feeding on the described metabolic pathway were part of the investigation. First, we hypothesized that N-reduced diets lead to changes in circulating AA concentrations that are detected by GCN2 and result in activation of the AAR pathway, as has already been observed in non-ruminants. In addition, we aimed to identify specific AA that might be responsible for triggering this signaling cascade. The second objective was to investigate the interaction effects of dietary N and P restriction. We expected the young goats to be deficient in P_i_ due to P-reduced feeding, which manifested as hypophosphatemia. P_i_ is essential for various cellular functions and the metabolism within the body. It is involved in the mechanism of energy transfer related to the formation of ATP, serves as a mediator of intracellular signaling, and regulates protein activity [20]. The addition or removal of a phosphate group to or from proteins acts like an on/off switch that alters the specific activity. This led to our second hypothesis that diet-induced hypophosphatemia has profound effects on triggering the AAR pathway and consequently the induction of FGF21. 

Although the aim of dietary restriction models is to reduce both N and P content in the diet, there are very few reports on the interaction between the effects of N and P reduction at the cellular level. There is a research gap here that should be closed by further studies. Our results are intended to illustrate how the AAR pathway responds to N- and/or P-reduced diets in ruminants, while highlighting the inherent complexity of dietary restriction models.

## 2. Results

### 2.1. Intake, Body Weight, and Daily Weight Gain

The results of feed intake are summarized in Table 1. They show that low-P feeding reduced the daily intake of dry matter (DM) and concentrate. The effects on weight in the different feeding groups are shown in Table 2. Daily body weight gain and final body weight of goats in the P-restricted feeding groups were significantly reduced. Feed efficiency was calculated as the difference between kg body weight gain and kg feed dry matter intake over the feeding period. P-reduction resulted in a significantly reduced average feed efficiency. N-reduced feeding had no effect on these parameters. The data on P-reduction were partially published by Behrens et al. [21]. 

### 2.2. Plasma Concentration of Urea, P_i_, and Calcium (Ca), Concentration of FGF21 in Serum, and Glucose Concentration in Whole Blood 

The plasma concentration of urea, P_i_, and Ca, as well as serum concentrations of FGF21 and whole blood glucose in young goats fed an N- and/or P-reduced diet are presented in Table 3. The LP diet resulted in a significant decrease in plasma urea concentration as a biomarker of overall protein/AA supply in the N-reduced feeding groups. This is reflected in a positive correlation (r = 0.832, *p* < 0.0001) between N-intake and plasma urea. Dietary P-restriction resulted in a significantly reduced plasma concentration of P_i_. Blood plasma calcium (Ca) concentration was significantly increased in the P-reduced feeding group, whereas double N and P restriction resulted in only an increasing trend, as documented by a significant interaction between N- and P-reduced feeding. Blood glucose levels remained stable in the studied groups. A low protein intake markedly increased the serum FGF21 concentration, which was visible in the N-reduced feeding group.

### 2.3. Concentration of Essential and Non-Essential Amino Acids

An N- and/or P-reduced diet induced changes in circulating AA profiles in young goats (Figure 1 and Figure 2). All AA concentrations measured in plasma are shown in Table 4 and Table 5. During dietary protein restriction, most of the measured EAA, i.e., threonine (Thr), valine (Val), leucine (Leu), and lysine (Lys), were significantly decreased. Phenylalanine (Phe) was significantly increased. Methionine (Met), isoleucine (Ile), and histidine (His) did not change significantly in the study groups. In the case of Thr, the decreasing effect of N-reduction was not observed in the double restricted feeding group, which was reflected in a significant interaction between N- and P-restricted feeding. The total amount of EAA in the N-reduced feeding group showed an increasing trend; percentage-wise, it was 19.3% less than in the control group and 10.9% less than in the control group when both P and N were reduced. The opposite effect was observed in the concentrations of NEAA when protein intake was reduced. In particular, glutamic acid (Glu), glycine (Gly), alanine (Ala), tyrosine (Tyr), and proline (Pro) were significantly increased in the N-restricted feeding groups. Serine (Ser) showed an increasing trend, whereas arginine (Arg) showed a decreasing trend. Glutamine (Gln) increased in the N-reduced feeding group and decreased in the P-reduced feeding group. Asparagine (Asn) did not change significantly in the study groups. The total amount of NEAA tended to increase in the N-reduced feeding groups; percentage-wise, this increase was 16.8% more than in the control group and 12.0% more than in the control group when both P and N were reduced.

### 2.4. Hepatic Expression of GCN2, ATF4, FGF21-mRNA

To investigate the cause of the increased circulating serum FGF21 level, the expression of genes in the liver involved in the AAR pathway was determined after feeding the young goats with an N- and/or P-reduced diet (Table 6). RT-PCR data were normalized by *18S* ribosomal RNA and *RPL19*. The hepatic mRNA expression of *GCN2* was significantly increased in the N-reduced feeding group. No changes in the gene expression of *GCN2* were detected in the P-reduced feeding group. Interestingly, no changes were observed in the N- and P-double-reduced feeding group either. This indicated an interaction effect between dietary P- and N-reduction. Moreover, similar results were obtained by quantification of *ATF4*-mRNA, which showed significantly increased expression in the N-reduced feeding group and no change in gene expression in the P-reduced and N- and P-double-reduced feeding group. Finally, hepatic *FGF21* mRNA expression increased highly significantly in animals receiving an N-reduced diet. *FGF21* gene expression remained unchanged in the P-reduced feeding group and in the N- and P-double-reduced feeding group. Figure 3 shows the results of two-way ANOVA of FGF21 in serum (a) and *FGF21* mRNA expression in liver (b) in comparison. The graphic picture that emerges is almost identical.

### 2.5. Hepatic Expression of GCN2—Protein

The results of the hepatic GCN2 protein expression level are presented in Table 7. The protein expression was significantly increased in the N-reduced feeding group. No changes in gene expression of GCN2 were observed in the P-reduced feeding group, nor in the N- and P-double-reduced feeding group (Figure 4a,b).

In the following, a comprehensive correlation analysis was performed between circulating plasma EAA and NEAA and plasma urea, P_i_, serum FGF21, *GCN2*, *ATF4*, *FGF21* mRNA expression, and GCN2 protein expression. The results are graphically presented in Figure 5. The *p*-values for linear regression are shown in Appendix A. Numerous significant correlations demonstrated the effects of the AA profile on the AAR signaling pathway and confirmed an association as discussed below.

## 3. Discussion

### 3.1. Amino Acids and Amino Acid Response Pathway

Several studies in different species have addressed the influence of LP diets on free plasma EAA and NEAA. Traditionally, EAA are those that are not synthesized in eukaryotes, whereas NEAA are synthesizable de novo in animal cells. Although the National Research Council (NRC) has recommended requirements of non-synthesizable AA for small ruminants [22], there is a lack of full understanding of AA biochemistry in the organisms, and the optimal dietary requirements are not decisively classifiable. Determining EAA and NEAA requirements for ruminants is particularly difficult compared to monogastric animals due to the complexity of N metabolism in the rumen and the ability of microorganisms to synthesize AA de novo [23,24]. The return of N from the gut to the body’s AA pool can account for up to 70% of dietary N intake and play an important role in maintenance of AA homeostasis [25]. In this study, we follow the traditional classification of AA based on human and animal nutrient requirements [26,27]. 

However, similar changes in the AA profile were observed in different animal species fed an LP diet. The decrease in EAA and increase in NEAA observed in this study when dietary N intake was restricted are in line with previous findings, e.g., in humans [28,29] and rats [30]. Laeger et al. analyzed the effects of dietary protein restriction on hepatic AA metabolism in rats. The changes to the AA profile are partially congruent with those observed in this study. In particular, the EAA Thr was significantly decreased, whereas some NEAA (Ala, Asp, Gly, Gln, Ser, and Tyr) were significantly increased (supplemental data [31]). The studies in ruminants on blood AA profile during dietary protein restriction are scarce. Gebeyew et al. found that feeding young goats with an LP diet decreases EAA His and Val, whereas it increases NEAA Gly [32]. Gly is one of the NEAA that was also increased in this study under protein restriction. It can be reversibly converted to Ser and both are metabolically important for protein synthesis and the synthesis of glutathione, phospholipids, nucleotides, and other metabolites [33]. The increase in this AA can be considered as an adaptive response to an LP diet to maintain important cellular processes. Regarding the reduction of EAA, it should be noted that AA supply depends not only on the total amount, but also on the quality of the dietary protein, or a protein source with a balanced AA composition. The consumption of a diet deficient in a single EAA results in a rapid and significant decrease in circulating levels of the missing EAA [34]. In yeast, the limitation of a single AA at the cellular level leads to activation of GCN2 as a primary response to nutritional stress [35]. The AAR pathway is generally thought to be activated to recognize and respond to AA deficiency [9]. The reduction in a specific AA to induce an AAR pathway in vivo or in vitro is an experimental model that has been used extensively to characterize the cellular transcriptional response to nutritional stress. The restriction of His [10], the branched chain AA (BCAA) Leu [36,37,38,39], and the sulfur-containing AA cysteine (Cys) [40], among many other examples, have been demonstrated to induce the AAR pathway. The reduction in AA supply leads to the accumulation of uncharged transfer ribonucleic acid (tRNA) via deacetylation [35], which binds GCN2 to initiate the AAR pathway. Our study of young goats fed an N-reduced diet revealed increased expression levels of genes involved in the AAR pathway, a finding demonstrated by other authors in non-ruminants, as described above. This shows that despite digestive peculiarities, a depletion of cellular AA also occurs in ruminants on reduced protein diets, as detected by GCN2. Figure 5 shows a visualized heatmap of Pearson’s correlation coefficient linking free EAA and NEAA to genes involved in the AAR pathway. In addition, the correlation between AA levels and plasma urea and plasma P_i_ concentrations as a result of dietary restriction are presented. Most of the circulating AA correlated either negatively or positively with blood parameters and genes (Appendix A). The EAA correlated predominantly negatively, while the NEAA correlated predominantly positively with genes involved in the AAR pathway. EAA, particularly BCAAs (Val, Iso, Leu), Thr, and Lys, correlated positively and obviously with the plasma urea levels, indicating that these AA levels responded well to dietary protein intake. The same EAA showed negative correlations with GCN2 and ATF4, indicating a comprehensible relationship with the induction of the AAR pathway. 

Negative correlations of urea with Gly, Ala, Tyr, and Pro as NEAA highlight the increase in these AA upon dietary protein reduction, which can be explained by the fact that a variety of ATF4-related genes are involved in AA metabolism and transport [11,12]. One of the functional consequences of ATF4 expression is to maintain intracellular AA levels by mediating the transcription of genes in response to low concentrations of cellular AA. For instance, ATF4-induced increases in the expression of enzymes involved in serine-glycine synthesis such as phosphoglycerate dehydrogenase (PHGDH), phosphoserine aminotransferase 1 (PSAT1), phosphoserine phosphatase (PSPH), and serine hydroxymethyltransferase (SHMT) 2 have been reported (Figure 6) [41]. Noguchi et al. examined dietary protein-dependent AA metabolism by linking free AA to transcriptional profiles in a protein restricted rat model. The plasma Ser level increased in response to decreases in protein intake, which was positively correlated with Phgdh expression [42]. Shmt 2 reversibly converts Ser into Gly. Both are required to support the biosynthesis of proteins, lipids, and nucleotides [43]. 

Moreover, ATF4 is described as a transcriptional activator of the alanine aminotransferase 2 (ALT2) gene, which catalyzes the reversible transamination between l-Ala and 2-oxoglutarate to pyruvate and l-Glu [44]. Tyr is biosynthesized from Phe via phenylalanine hydroxylase (PAH) [45]. Our results indicate that Phe is one of the EAA that is increased under dietary protein restriction, which could explain the increase in Tyr. Additionally, the 6-pyruvoyl-tetrahydropterin synthase (PTS) gene is activated by ATF4 and is as an important cofactor for hepatic PAH involved in the Phe-Tyr-pathway [46]. Finally, in the case of Pro, intracellular Pro concentrations have been shown to increase in response to ATF4 [47,48,49].

### 3.2. Amino Acids and FGF21

Another target gene for ATF4, related to dietary protein restriction, is the hormone FGF21. The first indication that FGF21 is also elevated in ruminants during dietary protein restriction was provided by Firmenich et al. [19]. As an important metabolic regulator in response to nutritional signals, it is induced in the liver by elevated levels of ATF4 and released into the bloodstream [13]. FGF21 was originally described as an adaptive response to starvation [17]. However, recent evidence indicates that FGF21 is induced by dietary protein restriction or deficiency of certain AA [38,50,51]. Circulating FGF21 levels are dramatically increased in response to LP diets in mice, rats, and humans [31]. As hypothesized, *FGF21* expression was strongly increased in young goats fed an LP diet, an effect associated with an elevation in blood FGF21 concentration. This outcome is revealed in a strongly positive correlation between the *FGF21* mRNA level and FGF21 in blood (Figure 7). Wanders et al. demonstrate that dietary Leu restriction increases FGF21 expression particularly in the liver, and additionally increases the hepatic release of FGF21 into the bloodstream [36]. The heatmap clearly shows that Leu as well as Thr, Val, Iso, and Lys are negatively correlated with FGF21. These AA are thought to play an important role in the initiation of the AAR pathway in ruminants. Our results lead to the conclusion that the increased hepatic *FGF21* mRNA expression and blood FGF21 levels are induced by AA deficiency due to protein-reduced feeding, and are mediated by the AAR signaling pathway.

### 3.3. Interaction of Low N and Low P Diet

In this study, the interactions between dietary N restriction and concurrent dietary P restriction were also investigated. From the heatmap (Figure 5), it can be seen that a correlation coefficient close to zero indicates a weak relationship between plasma P_i_ level and AA levels in blood. However, in this study, the relationship between P and N reduction is by no means irrelevant. A remarkable finding is that simultaneous reduction in N and P neither activates the AAR pathway nor induces *FGF21*. This metabolic adaptation in the animals complicates the interpretation of the experimental results. 

First, blood concentrations of EAA could increase as a result of protein breakdown supporting gluconeogenesis [52]. The effect of general dietary restriction, including caloric restriction, on profiles of free AA in plasma of goats could result in higher concentrations of EAA such as Val, Iso, Leu, and Thr [53], which could explain why the AAR pathway is not induced. In our study, there was no decrease in blood glucose concentration (Table 3), indicating that the animals did not have a negative energy balance. 

However, feeding a low P ration to young goats results in changes in mineral homeostasis, leading to pronounced hypophosphatemia and associated hypercalcemia (Table 3). These variations in plasma P_i_ levels could have a major impact. Marked P deprivation experimentally induced over four weeks in mice resulted in significant adenosine triphosphate (ATP) depletion, which could be reversed by P supplementation [54]. The energy released by the hydrolysis of ATP is used for numerous energy-requiring cellular reactions. The metabolic system for protein synthesis is usually the most energy-demanding process in organisms, and several observations indicate that this process is highly dependent on P availability [54,55,56]. Hence, under the condition of low P intake, protein synthesis and growth are reduced as a result of lower ATP availability [54,55], and AA eventually accumulate in the bloodstream. Panskeep and Booth suggested that hypothalamic AA sensing may contribute to a hypophagic response [57]. Reduced feed intake prevents excessive accumulation of AA in the bloodstream, which can lead to toxicity [58]. A reduction in feed intake and body weight gain induced by dietary P restriction has been reported by several authors [59,60], and was also observed in the young goats in the current study (Table 1 and Table 2). Due to the reduction in the rate of protein synthesis, circulating AA remain at a relatively constant level even when dietary protein intake is reduced (Figure 8). 

Special attention in this context is given to Thr, which is among the significantly reduced EAA in the N-reduced feeding group and remains at the same level as in the control group when goats were fed N- and P-restrictive diets simultaneously. Studies have demonstrated that deprivation of Thr was sufficient to mimic the effects of dietary EAA restriction with increased levels of serum FGF21 in mice [61]. Therefore, Thr can be considered to play an important role in the AAR pathway. The additional P reduction prevents a decrease in AA responsible for the initiation of the AAR pathway.

## 4. Materials and Methods

### 4.1. Animals, Feeding Regimen

Pertaining to the protection of experimental animals, all procedures with living goats used in this study were approved by the Animal Welfare Commissioner of the University of Veterinary Medicine Hannover (Hannover, Germany) and observed the German Animal Welfare Law (protocol code 33.19-42502-04-19/3076; 15 March 2019). In total, 28 male Colored German Goats entered the study with an initial weight of 19.04 (SEM 0.08) kg and were divided into 4 feeding groups with the following content of crude protein (CP) and phosphorus (P) in percentage of dry matter (DM): control diet (16.48% CP; 0.48% P; n = 7), N-reduced diet (8.35% CP; 0.51% P; n = 7), P-reduced diet (16.86% CP; 0.11% P; n = 7), and N- and P-reduced diet (8.1% CP; 0.11% P; n = 7). Each animal was fed 55 g pelleted concentrate per kg metabolic body size. Individual feeding was performed at 08.00 a.m. and 16.00 p.m. daily over a six-week period. Additionally, wheat straw was offered in group feeding (25% of the concentrate weight) and water was available ad libitum. The animals were weighed weekly, and their amount of feed was adjusted to the determined weight.

### 4.2. Diets

The pelleted feed was produced by a feed manufacturer specializing in the production of feed for animal research (ssniff Spezialdiäten GmbH, Soest, Germany). The composition of concentrate feed (DM, CP, crude ash CA, crude fat CL, crude fiber CF) was determined by the established Weender analysis, as per suggestions provided by the Association of German Agricultural Investigation and Research Center (VDLUFA). Neutral detergent fiber (NDF) and acid detergent fiber (ADF) were analyzed by the van Soest method and subsequently corrected to organic matter (ADFom; aNDFom) [62]. The diets were formulated isoenergetically with an energy level of 12.9 MJ ME/kg DM. The ingredients and compositions of concentrate feed and wheat straw are presented in Table 8. Daily N supply of the control- and P-reduced feeding groups and P supply of the control- and N-reduced feeding groups, as well as energy and Ca supply of all groups were based on recommendations of the Society of Nutrition Physiology (GfE) for young ruminating goat kids [63].

### 4.3. Blood and Tissue Sampling

After six weeks, the experimental feeding period ended with the slaughter of one animal per day from an alternating group by exsanguination following a standard abattoir captive bolt stunning procedure (Annex IV Directive 2010/63/EU). Blood samples of each individual animal were taken shortly before the slaughter by jugular venipuncture and collected in lithium heparin- and EDTA-coated syringes and serum syringes (Sarstedt AG & Co. KG, Nümbrecht, Germany). Cells were separated from whole blood by centrifugation at 2000× *g* for 15 min at room temperature (RT). Serum and plasma aliquots were stored at −20 °C. Tissue samples were taken from the liver within 5 min postmortem, rinsed with ice-cold saline (0.9% *w*/*v*), frozen in liquid N_2_, and stored at −80 °C until further preparation.

### 4.4. Biochemical Determinations

Changes in plasma urea levels as a direct effect of dietary protein intake were measured through a commercial assay (R-Biopharm AG, Pfungstadt, Germany). The analysis of P_i_ and calcium (Ca) concentrations in plasma was performed colorimetrically by conventional spectrometric methods. Glucose concentrations in fresh whole blood were measured by the glucose dehydrogenase method (Accu-Chek Performa, Roche Diabetes Care Austria GmbH, Wien, Austria). The serum FGF21 concentrations were assayed by a commercially available mouse/rat specific enzyme-linked immunosorbent assay (ELISA), for which a cross-reactivity for goat specific FGF21 was observed (BioVendor, Brno. Czech Republic; Cat. No. RD291108200R). Using an AA analyzer, the serum AA profile was determined by ion exchange chromatography (Biotronic LC 3000, Eppendorf, Maintal) [64]. The levels of Thr, Val, Met, Ile, Leu, Phe, Lys, and His as EAA and Arg, Ser, Asp, Glu, Gln, Gly, Ala, Tyr, and Pro as NEAA were measured.

### 4.5. RNA Isolation and Reverse Transcription

The RNeasy Mini Kit (Qiagen GmbH, Hilden, Germany), which integrates genomic DNA Eliminator spin columns with RNeasy technology, was used to isolate and purify total RNA from liver tissue. Analyses of RNA quantity were performed spectrophotometrically using NanoDrop One (Thermo Fisher Scientific Inc., Waltham, MA, USA). The RNA quality was assessed by RNA integrity numbers (RIN) by using an RNA 6000 nanoassay for an Agilent 2100 Bioanalyzer (Agilent Technologies Deutschland GmbH, Waldbronn, Germany). A reverse transcription of the isolated RNA was performed using the TaqMan™ Reverse Transcription Reagent (Applied Biosystems Inc., Thermo Fisher Scientific, Waltham, MA, USA) including a blend of oligo-dT primers and random hexamers to produce a global complementary DNA (cDNA) population from all transcripts in the RNA sample. The cDNA was then analyzed by a qPCR step, and gene-specific PCR-primer helped to determine the sequences of interest.

### 4.6. Hepatic Expression of GCN2, ATF4 and FGF21-mRNA

The resulting cDNA samples were used to amplify the target genes *GCN2*, *ATF4*, and *FGF21* and the reference genes *18S*, *B2M*, *RPS9*, *TBP*, and *RPL19* by quantitative real-time PCR (qPCR) in accordance with standard protocols. The gene-specific TaqMan primers for the reference gene *18S* were synthesized by TIB MOLBIOL (Berlin, Germany; Table 9). The reaction mixes of 20 µL contained TaqMan™ Gene Expression Master Mix (Applied Biosystems Inc., Thermo Fisher Scientific Inc.), 16 ng reverse transcripted cDNA, 300 nmol/L specific primers and 100 nmol/L of each sample. Using a real-time PCR cycler (CFX96^TM^, Bio-Rad Laboratories GmbH, Feldkirchen, Germany), PCR products were amplified under the following reaction conditions: 50 °C for 2 min, 95 °C for 10 min, 40 cycles at 95 °C for 15 s, and 60 °C for 1 min. For *GCN2*, *ATF4*, *FGF21*, *RPL19*, *B2M*, *RPS9*, and *TBP*, gene-specific primers were synthesized by Thermo Fisher Scientific Inc. To quantify levels of gene expression, we used SYBR Green^®^ PCR assays with specific primers (Table 10). Reaction mixtures of 20 µL contained SensiFAST^TM^ SYBR No-Rox Mix (BioCat GmbH, Heidelberg, Germany), 200 nmol/L specific primers, and 16 ng reverse transcribed cDNA. The qPCR cycling conditions were as follows: 3 min at 95 °C, followed by 40 cycles at 95 °C for 10 sec, and 30 sec at 60 °C (63 °C in case of *FGF21*) using a real-time PCR cycler (CFX96^TM^, Bio-Rad). The melt curve protocol was followed with an incubation for 10 min at 55 °C and then 10 s each at 0.5 °C increments up to 95 °C. The absolute copy number of the target gene in the sample was calculated in reference to a generated standard curve. The verification of amplicon specificity was performed by sequencing (Microsynth Seqlab GmbH, Göttingen, Germany) and using NCBI Blast (Bethesa, MD, USA). *18S* and *RPL19* were considered to be the best pair of reference genes for normalization using NormFinder software v0.953, available online: https://www.moma.dk/normfinder-software (accessed on 1 April 2022).

### 4.7. Hepatic Expression of GCN2—Protein

To quantify hepatic protein expression level of GCN2 via the Western blot analysis method, we performed a cell lysate preparation. The frozen hepatic tissues were mechanically homogenized in buffer on ice containing 150 mM NaCl, 1% Nonidet, 5 mM EDTA, and 50 mM Tris at pH 8.0 using a potter-homogenizer (B. Braun Melsungen AG, Melsungen, Germany). After an incubation time of 45 min at 4 °C, samples were centrifuged at 12,000× *g* for 30 min at 4 °C. The supernatant was removed in a new tube and a protease and phosphatase inhibitor were added. Samples were stored in aliquots at −20 °C. The commonly used Bradford Assay (Serva Elektrophoresis GmbH, Heidelberg, Germany) was the preferred method for measuring the protein concentration. Bovine serum albumin (BSA) was measured as a standard along with the analysis of each unknown sample (Catalog # A-2153, Sigma-Aldrich, St. Louis, MO, USA). To evaluate the protein expression level, 25 µg of protein were loaded on 6% SDS-PAGE–polyacrylamide gels and were separated through gel electrophoresis, followed by electroblotting of proteins onto nitrocellulose membranes. The blocking of non-specific binding was achieved by placing the membrane in PBS-Tween^®^ with 5% non-fat dried milk for 1 h. The blots were incubated at 4 °C overnight with specific primary antibody. The primary antibodies included anti-GCN2 polyclonal antibody (Catalog # 720463, Thermo Fisher Scientific) diluted 1:2500 in PBS-Tween^®^. Enzyme-linked immunodetection of antigen-specific antibodies was performed by using anti-rabbit secondary antibodies (diluted 1:20,000 in 5% milk/PBS-Tween^®^) conjugated with horseradish peroxidase (HRP). The targeted proteins were visualized with Pierce SuperSignal West (Thermo Fisher Scientific) and imaged on ChemiDoc™ MP Imaging System (Bio-Rad). Subsequently, a densitometric analysis of proteins was performed using the analysis software ImageLab 5.2.1 (Bio-Rad). The target proteins were normalized to the total amount of protein in each lane.

### 4.8. Statistical Analysis

Statistical analyses and visualization were performed using GraphPad Prism software version 9.3 (GraphPad Software, San Diego, CA, USA). Normality was assessed by the Shapiro–Wilk test, and the data were analyzed by two-way ANOVA followed by the post hoc Tukey test for multiple comparison. The standard error of the mean (SEM) was used to express the variability of data. Differences were considered to be significant at *p* < 0.05. Pearson’s correlation was used to indicate how strongly two variables were linearly related. The correlation matrix heatmap was used to visualize data correlations to uncover possible interactions between AA and partners of the AAR pathway, and to provide an overview of trends and significances.

## 5. Conclusions

In summary, the contribution of this study is, first, that the LP diet alters the AA profile in the blood of young goats despite their complexity of rumen N metabolism, leading to initiation of the GCN2/eIF2α/ATF4 pathway. Moreover, the pathway described was determined as a link between reduced protein intake and increased hepatic *FGF21* expression and circulating FGF21. Second, activation of the GCN2/eIF2α/ATF4 pathway is dependent on the adequate availability of P. When P availability is low, the AAR pathway and FGF21 are not induced.

## Figures and Tables

**Figure 1 ijms-24-07153-f001:**
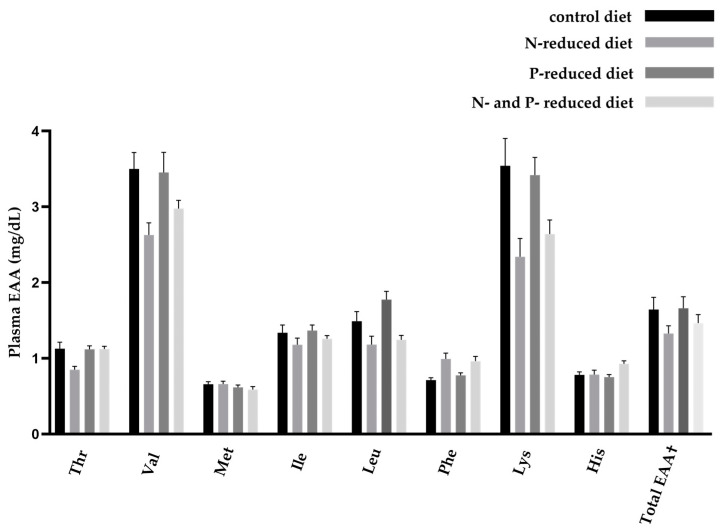
Overview of circulating essential amino acids (EAA) in young goats fed an N- and/or P-reduced diet. Mean values with their standard errors. Mean values that differ significantly in the Tukey test are shown in Table 4 with different superscript letters. n = 7 animals. † n = 56 animals.

**Figure 2 ijms-24-07153-f002:**
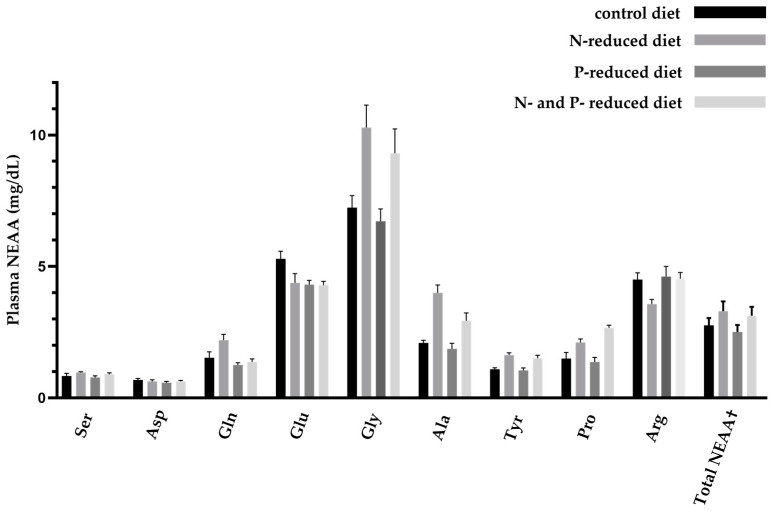
Overview of circulating non-essential amino acids (NEAA) in young goats fed an N- and/or P-reduced diet. Mean values with their standard error. Mean values that differ significantly in the Tukey test are shown in Table 5 with different superscript letters. n = 7 animals. † n = 63 animals.

**Figure 3 ijms-24-07153-f003:**
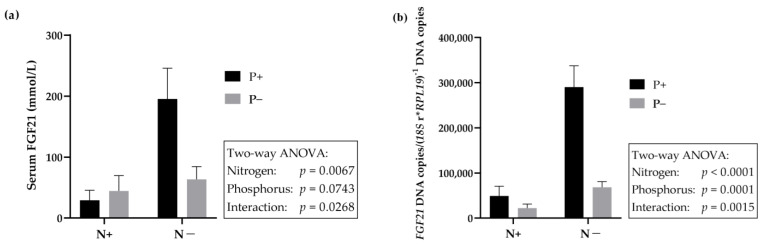
(**a**) Serum FGF21 and (**b**) hepatic *FGF21* mRNA expression normalized to *18S*/*RPL19* in young goats fed an N- and/or P-reduced diet. Two-way ANOVA analysis; *p* < 0.05.

**Figure 4 ijms-24-07153-f004:**
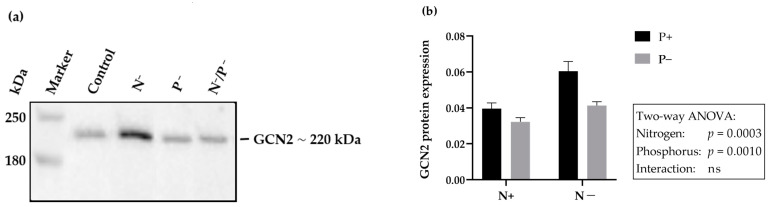
(**a**) GCN2 protein expression in the liver of young goats fed an N- and/or P-reduced diet, determined by Western blotting. Expression of GCN2 was quantified by densitometry and normalized to the total amount of protein. Western blotting results showed a representative blot. (**b**) Results were analyzed using two-way ANOVA analysis (*p* < 0.05), followed by post hoc Tukey test for multiple comparison.

**Figure 5 ijms-24-07153-f005:**
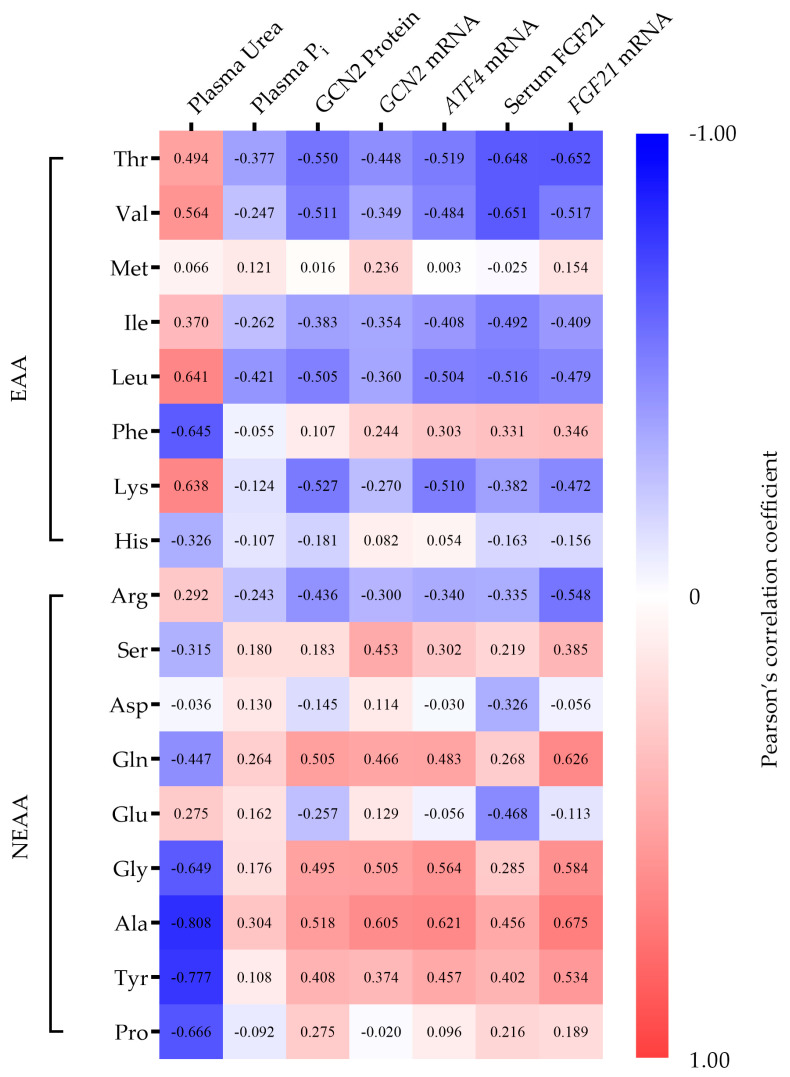
Correlation heatmap of circulating plasma EAA and NEAA and plasma urea, plasma P_i_, serum FGF21, *GCN2* mRNA, *ATF4* mRNA, *FGF21* mRNA expression, and GCN2 protein expression in young goats fed an N- and/or P-reduced diet. The dataset was formed by combining the control group (n = 7), N-reduced group (n = 7), P-reduced group (n = 7), and N- and P-reduced group (n = 7). The value of the correlation coefficient is indicated by the color range from 1 (strongly positive: deep red) to −1 (strongly negative: deep blue).

**Figure 6 ijms-24-07153-f006:**
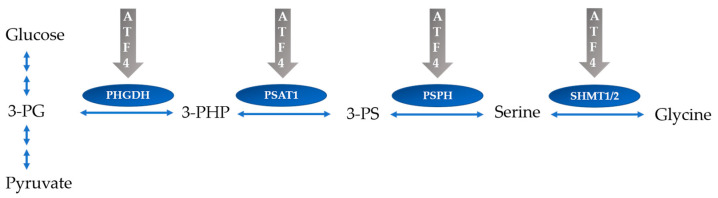
De novo serine–glycine biosynthesis. ATF4 leads to an increase in the expression of enzymes involved in serine–glycine synthesis. 3-PG is converted into serine and glycine by four enzymatic reactions. 3-PG (3-phosphoglycerate); PHGDH (phosphoglycerate dehydrogenase); 3-PHP (3-phosphohydroxypyruvate); PSAT1 (phosphoserine aminotransferase 1); 3-PS (3-phosphoserine); PSPH (phosphoserine phosphatase); SHMT1/2 (serine hydroxymethyltransferase 1/2).

**Figure 7 ijms-24-07153-f007:**
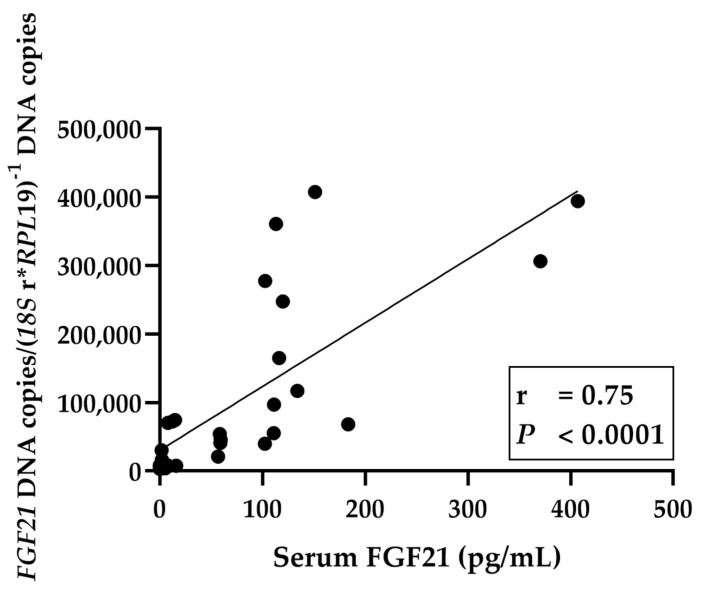
Linear regression of *FGF21* DNA expression with serum FGF21 in young goats fed an N- and/or P-reduced diet. The level of significance with Pearson’s correlation coefficient was set at *p* = 0.05.

**Figure 8 ijms-24-07153-f008:**
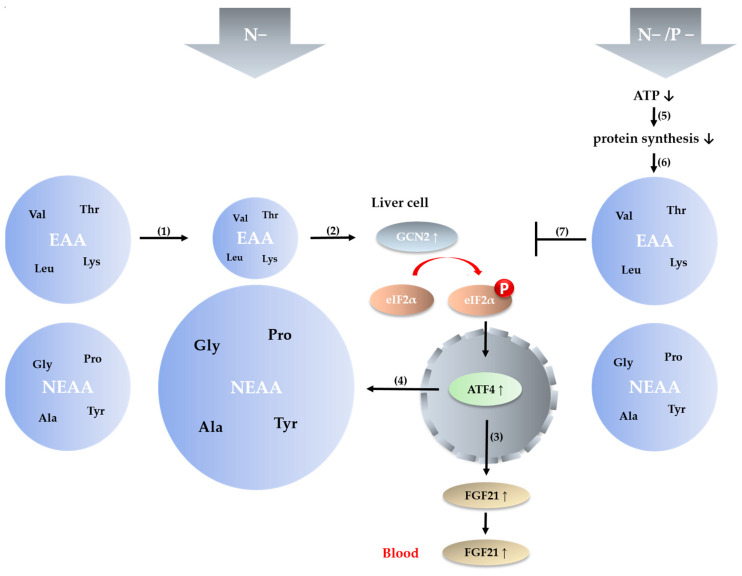
Dietary N reduction leads to depletion of EAA (1), which results in activation of the GCN2/eIF2α/ATF4 pathway (2) and the induction of FGF21 (3). Increased expression of ATF4-related genes leads to an increase in NEAA (4). Dietary P reduction leads to reduced protein synthesis via reduced availability of ATP (5). EAA are not reduced with simultaneous N reduction (6) and prevent the signaling pathway from being initiated (7).

**Table 1 ijms-24-07153-t001:** Mean daily intakes of DM, concentrate, N, and P of young goats fed an N- and/or P-reduced diet.

Items	N+/P+	N−/P+	N+/P−	N−/P−	*p*-Value		
					N-Reduction	P-Reduction	Interaction
DM intake (g/d)	614 ± 10 ^a^	594 ± 6 ^a,b^	528 ± 25 ^b^	534 ± 25 ^b^	0.707	0.001	0.485
Concentrate intake (g/d)	555 ± 12 ^a^	537 ± 7 ^a,b^	469 ± 29 ^b^	475 ± 28 ^a,b^	0.799	0.002	0.581
N intake (g/d)	13.39 ± 0.27 ^a^	6.76 ± 0.09 ^b^	11.64 ± 0.68 ^c^	5.87 ± 0.33 ^b^	<0.0001	0.003	0.308
P intake (g/d)	2.40 ± 0.05 ^a^	2.49 ± 0.03 ^a^	0.53 ± 0.03 ^b^	0.54 ± 0.03 ^b^	0.189	<0.0001	0.282
Ca intake (g/d)	6.50 ± 0.13 ^a^	6.20 ± 0.08 ^a,b^	5.42 ± 0.31 ^b,c^	5.22 ± 0.29 ^c^	0.274	0.0001	0.827

DM (dry matter), N (nitrogen), P (phosphorus). Mean values ± standard errors. n = 7 animals. ^a,b,c^ Mean values within a row with different superscript letters were significantly different; Tukey’s multiple comparisons test (*p* < 0.05).

**Table 2 ijms-24-07153-t002:** Initial body weight and effects on body weight gain, final body weight, and feed efficiency of young goats fed an N- and/or P-reduced diet.

Items	N+/P+	N−/P+	N+/P−	N−/P−	*p*-Value		
					N-Reduction	P-Reduction	Interaction
Initial body weight (kg) *	19.21 ± 0.75	18.86 ± 0.57	18.93 ± 1.09	19.14 ± 1.01	0.936	>0.9999	0.749
Final body weight (kg) †	25.29 ± 0.74 ^a^	24.21 ± 0.32 ^a,b^	20.21 ± 0.93 ^c^	21.14 ± 1.22 ^b,c^	0.935	<0.0001	0.261
Body weight gain (kg/d)	0.14 ± 0.01 ^a^	0.12 ± 0.02 ^a^	0.03 ± 0.01 ^b^	0.05 ± 0.01 ^b^	0.907	<0.0001	0.091
Feed efficiency (kg/kg DM)	0.23 ± 0.02 ^a^	0.21 ± 0.02 ^a^	0.05 ± 0.02 ^b^	0.09 ± 0.02 ^b^	0.822	<0.0001	0.118

Mean values ± standard errors. n = 7 animals. * The initial body weight was determined at the beginning of the experimental feeding at 10 weeks of age. † The final body weight was determined at the time of slaughter at 15–17 weeks of age. ^a,b,c^ Mean values within a row with different superscript letters were significantly different; Tukey’s multiple comparisons test (*p* < 0.05).

**Table 3 ijms-24-07153-t003:** Plasma concentration of urea, P_i_ and Ca, concentration of FGF21 in serum, and glucose concentration in whole blood of young goats fed an N- and/or P-reduced diet.

Items	N+/P+	N−/P+	N+/P−	N−/P−	*p*-Value		
					N-Reduction	P-Reduction	Interaction
Urea (mmol/L)	6.76 ± 0.46 ^a^	1.16 ± 0.09 ^b^	6.85 ± 0.18 ^a^	2.22 ± 0.66 ^b,c^	<0.0001	0.180	0.250
P_i_ (mmol/L)	2.09 ± 0.12 ^a^	2.00 ± 0.21 ^a^	0.70 ± 0.04 ^b^	1.06 ± 0.09 ^b,c^	0.301	<0.0001	0.099
Ca (mmol/L)	3.16 ± 0.09 ^a^	3.11 ± 0.09 ^a^	4.34 ± 0.17 ^b^	3.55 ± 0.12 ^a,c^	0.002	<0.0001	0.005
Glucose (mg/dL)	68.00 ± 2.66	67.86 ± 1.10	67.43 ± 1.67	66.57 ± 2.55	0.814	0.662	0.866
FGF21 (mmol/L)	29.14 ± 16.43 ^a^	195.3 ± 50.51 ^b^	44.51 ± 25.16 ^a^	63.53 ± 20.86 ^a^	0.007	0.074	0.027

P_i_ (inorganic phosphate), Ca (calcium), FGF21 (fibroblast growth factor 21). Mean values ± standard errors. n = 7 animals. ^a,b,c^ Mean values within a row with different superscript letters were significantly different; Tukey’s multiple comparisons test (*p* < 0.05).

**Table 4 ijms-24-07153-t004:** Circulating EAA concentration in young goats fed an N- and/or P-reduced diet.

Items	N+/P+	N−/P+	N+/P−	N−/P−	*p*-Value		
					N-Reduction	P-Reduction	Interaction
Threonine (mg/dL)	1.13 ± 0.08 ^a^	0.85 ± 0.04 ^b^	1.12 ± 0.05 ^a^	1.13 ± 0.03 ^a^	0.022	0.026	0.018
Valine (mg/dL)	3.50 ± 0.22 ^a^	2.63 ± 0.16 ^b^	3.45 ± 0.26 ^a^	2.98 ± 0.11 ^a,b^	0.002	0.447	0.325
Methionine (mg/dL)	0.66 ± 0.03	0.66 ± 0.04	0.62 ± 0.03	0.59 ± 0.04	0.734	0.130	0.679
Isoleucine (mg/dL)	1.34 ± 0.10	1.18 ± 0.09	1.37 ± 0.07	1.26 ± 0.04	0.105	0.504	0.766
Leucine (mg/dL)	1.49 ± 0.13 ^a,c^	1.18 ± 0.11 ^a^	1.78 ± 0.11 ^c^	1.24 ± 0.06 ^a^	0.0005	0.111	0.299
Phenylalanine (mg/dL)	0.71 ± 0.03 ^a^	0.99 ± 0.08 ^b^	0.78 ± 0.03 ^a,c^	0.96 ± 0.06 ^b,c^	0.0003	0.754	0.414
Lysine (mg/dL)	3.54 ± 0.36 ^a^	2.34 ± 0.24 ^b^	3.42 ± 0.23 ^a^	2.64 ± 0.19 ^a,b^	0.0009	0.739	0.428
Histidine (mg/dL)	0.78 ± 0.04 ^a,b^	0.79 ± 0.06 ^a,b^	0.75 ± 0.03 ^b^	0.93 ± 0.04 ^a^	0.218	0.056	0.065
Total EAA † (mg/dL)	1.65 ± 0.16	1.33 ± 0.10	1.66 ± 0.15	1.47 ± 0.11	0.058	0.566	0.650

EAA (essential amino acids). Mean values ± standard errors. n = 7 animals. † n = 56. ^a,b,c^ Mean values within a row with different superscript letters were significantly different; Tukey’s multiple comparisons test (*p* < 0.05).

**Table 5 ijms-24-07153-t005:** Circulating NEAA concentration in young goats fed an N- and/or P-reduced diet.

Items	N+/P+	N−/P+	N+/P−	N−/P−	*p*-Value		
					N-Reduction	P-Reduction	Interaction
Arginine (mg/dL)	4.50 ± 0.26	3.57 ± 0.18	4.61 ± 0.39	4.53 ± 0.24	0.081	0.063	0.138
Serine (mg/dL)	0.83 ± 0.10	0.97 ± 0.03	0.78 ± 0.07	0.89 ± 0.06	0.089	0.378	0.919
Asparagine (mg/dL)	0.68 ± 0.06	0.63 ± 0.06	0.58 ± 0.05	0.63 ± 0.04	0.960	0.330	0.321
Glutamic acid (mg/dL)	1.53 ± 0.22 ^a,b^	2.19 ± 0.22 ^b^	1.25 ± 0.09 ^a^	1.36 ± 0.12 ^a^	0.035	0.004	0.126
Glutamine (mg/dL)	5.29 ± 0.28 ^a^	4.38 ± 0.35 ^a,b^	4.31 ± 0.16 ^b^	4.29 ± 0.14 ^b^	0.074	0.043	0.085
Glycine (mg/dL)	7.24 ± 0.46 ^a^	10.28 ± 0.86 ^b,c^	6.72 ± 0.47 ^a^	9.31 ± 0.93 ^a,c^	0.0006	0.303	0.754
Alanine (mg/dL)	2.09 ± 0.10 ^a,c^	4.00 ± 0.30 ^b^	1.87 ± 0.21 ^c^	2.94 ± 0.30 ^a^	<0.0001	0.014	0.097
Tyrosine (mg/dL)	1.09 ± 0.05 ^a^	1.63 ± 0.09 ^b^	1.05 ± 0.09 ^a^	1.51 ± 0.11 ^b^	<0.0001	0.352	0.675
Proline (mg/dL)	1.50 ± 0.23 ^a,c^	2.11 ± 0.13 ^a,b^	1.36 ± 0.18 ^c^	2.67 ± 0.09 ^b^	<0.0001	0.215	0.047
Total NEAA † (mg/dL)	2.75 ± 0.29	3.31 ± 0.37	2.50 ± 0.27	3.13 ± 0.34	0.066	0.504	0.914

NEAA (non-essential amino acids). Mean values ± standard errors. n = 7 animals. † n = 63 animals. ^a,b,c^ Mean values within a row with different superscript letters were significantly different; Tukey’s multiple comparisons test (*p* < 0.05).

**Table 6 ijms-24-07153-t006:** Relative amounts of *GCN2*, *ATF4*, and *FGF21*-mRNA level in the livers of young goats fed an N- and/or P-reduced diet.

Items	N+/P+	N−/P+	N+/P−	N−/P−	*p*-Value		
					N-Reduction	P-Reduction	Interaction
*GCN2*	3.86 × 10^4^ ± 0.32 × 10^4 a^	5.50 × 10^4^ ± 0.64 × 10^4 b^	3.55 × 10^4^ ± 0.20 × 10^4 a^	3.89 × 10^4^ ± 0.36 × 10^4 a,b^	0.025	0.029	0.129
*ATF4*	9.21 × 10^6^ ± 0.48 × 10^6 a^	15.15 × 10^6^ ± 1.15 × 10^6 b^	10.15 × 10^6^ ± 1.14 × 10^6 a^	11.74 × 10^6^ ± 1.30 × 10^6 a,b^	0.002	0.259	0.052
*FGF21*	4.92 × 10^4^ ± 2.14 × 10^4 a^	29.03 × 10^4^ ± 4.74 × 10^4 b^	2.21 × 10^4^ ± 0.91 × 10^4 a^	6.82 × 10^4^ ± 1.25 × 10^4 a^	<0.0001	0.0001	0.002

*GCN2* (general control nonderepressible 2), *ATF4* (activating transcription factor 4). Expression levels of target genes were normalized to *18S*/*RPL19*. Mean values ± standard errors. n = 7 animals. ^a,b^ Mean values within a row with different superscript letters were significantly different; Tukey’s multiple comparisons test (*p* < 0.05).

**Table 7 ijms-24-07153-t007:** Relative amount of GCN2 protein expression in the liver of young goats fed an N- and/or P-reduced diet.

Items	N+/P+	N−/P+	N+/P−	N−/P−	*p*-Value		
					N-Reduction	P-Reduction	Interaction
GCN2	3.96 × 10^−2^ ± 0.32 × 10^−2 a^	6.04 × 10^−2^ ± 0.55 × 10^−2 b^	3.23 × 10^−2^ ± 0.23 × 10^−2 a^	4.13 × 10^−2^ ± 0.21 × 10^−2 a,b^	0.025	0.029	0.129

Mean values ± standard errors; n = 7 animals. ^a,b^ Mean values within a row with different superscript letters were significantly different; Tukey’s multiple comparisons test (*p* < 0.05).

**Table 8 ijms-24-07153-t008:** Components and composition of wheat straw and pelleted concentrate diets.

Items	Wheat Straw	Control	N-Reduction	P-Reduction	N- and P-Reduction
**Components (g/kg)**					
Soybean meal		68.0	57.0	68.0	57.0
Urea		24.5	-	24.5	-
Wheat starch		378	385	379	390
Beet pulp		399	415	399	410
Mineral-vitamin premix †		10.0	10.0	10.0	10.0
MgHPO_4_·3H_2_O		9.3	9.3	-	-
MgO		-	-	2.2	2.2
NaH_2_PO_4_·2H_2_O		9.7	10.0	0.4	0.9
NaCl		1.4	1.2	1.4	1.2
NaHCO_3_		-	-	5.0	5.0
CaCO_3_		14.3	14.2	14.3	14.2
Sipernat 22S ^‡^		41.8	58.1	52.3	69.5
Molasses		10.0	10.0	10.0	10.0
Soybean oil		34.0	30.0	34.0	30.0
**Composition ***					
DM (g/kg)	904	880	874	884	875
Nutrients (g/kg DM)					
Crude ash	39.8	100	112	101	115
Crude protein	25.4	165	83.5	169	81.1
ADFom	579	87.5	96.1	87.1	84.6
aNDFom	855	157	169	179	173
Crude fat	24.3	43.2	49.2	47.5	43.4
Urea	BDL	27.8	BDL	31.1	BDL
Ca	2.8	12.6	12.5	12.3	11.8
P	BDL	4.8	5.1	1.1	1.1
Vitamin D3 (IU/kg DM)	BDL	1000	1144	1075	1051
ME (MJ/kg DM)	8.3	12.8	12.4	12.9	12.5

BDL, below detection level; DM, dry matter; ME, metabolizable energy; ADFom, acid-detergent fiber expressed exclusive of residual ash; aNDFom, neutral-detergent fiber assayed with heat stable amylase expressed exclusive of residual ash. * Composition analyzed by the Association of German Agricultural Investigation and Research Center (VDLUFA). † Mineral-vitamin premix per kg: 0.2 g P; 12.1 g Ca; 1.7 g Na; 2.2 g Mg; 1,200,000 IU vitamin A; 120,000 IU vitamin D; 10,000 mg vitamin E; 675 mg vitamin K; 4960 mg iron; 6336 mg Zn; 501 mg Cu; 3000 mg Mn; 201 mg Co; 15 mg Se; 202 mg I. ^‡^ Sipernat 22S (Evonik Industries AG, Essen, Germany) is a fine particle silica used as an indigestible marker to adjust the weight of the N-reduced diet.

**Table 9 ijms-24-07153-t009:** Primers and probes used for TaqMan^TM^ assays.

Genes	Primers and Probes (5′ → 3′)	Accession Number	Reference
*18s*	Forward: AAAAATAACAATACAGGACTCTTTCG Reverse: GCTATTGGAGCTGGAATTACCG FAM-TGGAATGAGTCCACTTTAAATCCTTCCGC-BBQ	AM711869.1	[65]

**Table 10 ijms-24-07153-t010:** Primers used for SYBR green assays.

Genes	Primers and Probes (5′ → 3′)	Accession Number	Reference
*GCN2*	Forward: GAGACACCATTGACCAGGGG Reverse: TAGATGGTCGGTGGCCAAAC	XM_018054468.1 and XM_018054469.1	This study
*ATF4*	Forward: GTTCTCCTGCGACAAGGCTA Reverse: TGGCATGGTTTCCAGGTCAT	XM_018048794.1	This study
*FGF21*	Forward: CCTCTACACGGATGATGCCC Reverse: GCTTTGGGGTCAAAGTGCAG	XM_005692688.3	[19]
*RPL19*	Forward: AGCCTGTGACTGTCCATTCC Reverse: ACGTTACCTTCTCGGGCATT	XM_005693740.3	[66]
*B2M*	Forward: CCTTGGTCCTTCTCGGGCTG Reverse: TCTGGCGGGTGTCTTGAGTAT	XM_018053818.1	[66]
*RPS9*	Forward: CGCCTCGACCAAGAGCTGAAG Reverse: CTCCAGACCTCACGTTTGTTCC	XM_018063497.1	[67]
*TBP*	Forward: AGAAGGCCTTGTGCTAACCC Reverse: AGCAGCCATTACGTCGTCTT	XM_018053502.1 and XM_018053503.1	This study

## Data Availability

The data presented in the study are available on request from the corresponding author.

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
