# Peer review of "Modulation of GCN2/eIF2α/ATF4 Pathway in the Liver and Induction of FGF21 in Young Goats Fed a Protein- and/or Phosphorus-Reduced Diet"

_ijms, 2023, doi:10.3390/ijms24087153_

Round 1

Reviewer 1 Report

It is an effective approach to reduce nitrogen and P excretion by animals by reducing the content of crude protein (CP) and phosphorus (P) in the diets. However, a sufficient quantity and quality of protein in the diet, as well as an adequate amount of P, is necessary to meet nutritional requirements. A low-protein (LP) and/or low-phosphorus (P) diet may lead to a reduction in certain cellular EAA regardless of the species. This study intends to illustrate how the amino acid response (AAR) pathway responds to N- and/or P- reduced diets in ruminants, while highlighting the inherent complexity of dietary restriction models. Thus, although there have been many similar studies published, I believe that this one is interesting and of great value within the scope of the journal.   I trust, on the whole, that the study is well designed and well organized. Evaluation, methodology and statistical analysis are appropriate. The main conclusions are well supported by their data sets, and the logic lines of the manuscript run smoothly. Therefore, the overall merit of this manuscript is good and it provides helpful insights for the scientific community.   Here I would like to give the author some suggestions which I hope will contribute to the improvement of this manuscript.   First, I personally think the sections of introduction, discussion (especially discussion 3.1), conclusion are somewhat wordy and unconcise.   Second, in Figure 1, I wonder if the authors forget to label the significant difference markers. It appears that the data shown in Figure 1 are covered by Table 4, so the authors should consider if Figure 1 is necessary. The same holds for Figures 2 and Table 5, Figures 3 and Table 6, and Figures 5 and Table 7.   Third, to help the reader better understand the manuscript, the authors are encouraged to plot the diagrams of several mechanisms involved in molecular dynamics. In particular, in the discussion and conclusion sections, it follows that mechanism diagrams are required. This is an effective way to emphasize the conclusion by starting a new first class title.   In addition, I would like to stress that an objective summary of previous studies and pointing out research gaps is absolutely necessary in the introduction section. The author should therefore pay some attention to this point.   Correspondingly, I believe that this manuscript, with minor revisions, may be considered acceptable.

Author Response

Dear editor and dear reviewers,

Thank you very much for taking the time to review our manuscript entitled “Modulation of GCN2/eIF2α/ATF4 Pathway in the Liver and Induction of FGF21 in Young Goats Fed a Protein- and/or Phosphorus-reduced Diet”. We greatly appreciate your thoughtful comments that helped improve the manuscript. We have modified the manuscript according to the comments and suggestions. Revisions made to the manuscript are marked up using the “Track Changes” function. The changes in the manuscript are explained below.

It is an effective approach to reduce nitrogen and P excretion by animals by reducing the content of crude protein (CP) and phosphorus (P) in the diets. However, a sufficient quantity and quality of protein in the diet, as well as an adequate amount of P, is necessary to meet nutritional requirements. A low-protein (LP) and/or low-phosphorus (P) diet may lead to a reduction in certain cellular EAA regardless of the species. This study intends to illustrate how the amino acid response (AAR) pathway responds to N- and/or P- reduced diets in ruminants, while highlighting the inherent complexity of dietary restriction models. Thus, although there have been many similar studies published, I believe that this one is interesting and of great value within the scope of the journal.  

I trust, on the whole, that the study is well designed and well organized. Evaluation, methodology and statistical analysis are appropriate. The main conclusions are well supported by their data sets, and the logic lines of the manuscript run smoothly. Therefore, the overall merit of this manuscript is good and it provides helpful insights for the scientific community.  

Here I would like to give the author some suggestions which I hope will contribute to the improvement of this manuscript.  

First, I personally think the sections of introduction, discussion (especially discussion 3.1), conclusion are somewhat wordy and unconcise.  

Author: In some passages, the introduction and discussion were made more precise. The conclusion has been clarified and shortened.

Line 58, 84, 269, 273-274, 300, 429

Second, in Figure 1, I wonder if the authors forget to label the significant difference markers. It appears that the data shown in Figure 1 are covered by Table 4, so the authors should consider if Figure 1 is necessary. The same holds for Figures 2 and Table 5, Figures 3 and Table 6, and Figures 5 and Table 7.  

Author: You have correctly noted that in Figure 1 the significant difference markers are not labelled, although numerous significances were found in the Tukey test for multiple comparisons. However, to point these out would give a confusing picture. The decisive factor that will be shown here is that the trend of the change under protein reduction is the same for the essential amino acids on the one hand and the non-essential amino acids on the other. This is also the reason why the corresponding table is shown in addition to the figure. To avoid misunderstandings, we have added a reference to the table in the figure's footnote legend (Line 179, 191, “Mean values that differ significantly in the Tukey test are shown in Table 4/5 with different superscripts letters”). The combined presentation of Figures 3 and 5 and corresponding Tables increases the overview and clarifies relevant results.

Third, to help the reader better understand the manuscript, the authors are encouraged to plot the diagrams of several mechanisms involved in molecular dynamics. In particular, in the discussion and conclusion sections, it follows that mechanism diagrams are required. This is an effective way to emphasize the conclusion by starting a new first class title.  

Author: To underline the discussion and the conclusion and make it more comprehensible, we have included mechanism diagrams (Figure 7 and 8).

Line 341,

In addition, I would like to stress that an objective summary of previous studies and pointing out research gaps is absolutely necessary in the introduction section. The author should therefore pay some attention to this point.  Correspondingly, I believe that this manuscript, with minor revisions, may be considered acceptable.

Author: We have reviewed previous studies to the best of our knowledge. We have included a reference to research gaps in the introductory section (Line 92).

We would like to thank you once again for your helpful comments and suggestions for improving the manuscript, and hope that the change was in your interest.

Kind regards,

Sarah Weber

Reviewer 2 Report

The manuscript entitled “Modulation of GCN2/eIF2α/ATF4 Pathway in the Liver and Induction of FGF21 in Young Goats Fed a Protein- and/or Phosphorus-reduced Diet” is significant in this field of interest. The manuscript is well-structured with enough data. However, this manuscript has minor issues needed to be addressed. Thus I recommend this manuscript for minor revision. 

1.  All table titles are too long. Please shorten the title to fit into a single line and provide the remaining information as a table footnote/legend. For example: DM (dry matter), N (nitrogen), and P (phosphorus). Mean values±standard errors; n = 7 animals. a,b,c Mean values within a row with different superscript letters were significantly different; Tukey’s multiple comparisons test (p < 0.05).

2. The legends for Figures 1 and 2 should be moved above the graphs.

3. Tables 8, 9, 10, and 11 should be converted into supplementary tables.

4. The manuscript must be subjected to a typographical error check. For example, "concentrate feed und wheat straw" should be presented as "concentrate feed and wheat straw."

5. I suggest conducting a PCA analysis of the data to make it easier to understand the changes in EAA (essential amino acids) and LEAA (non-essential amino acids).

6. Kindly shorten the conclusion

Author Response

Dear editor and dear reviewers,

Thank you very much for taking the time to review our manuscript entitled “Modulation of GCN2/eIF2α/ATF4 Pathway in the Liver and Induction of FGF21 in Young Goats Fed a Protein- and/or Phosphorus-reduced Diet”. We greatly appreciate your thoughtful comments that helped improve the manuscript. We have modified the manuscript according to the comments and suggestions. Revisions made to the manuscript are marked up using the “Track Changes” function. The changes in the manuscript are explained below.

The manuscript entitled “Modulation of GCN2/eIF2α/ATF4 Pathway in the Liver and Induction of FGF21 in Young Goats Fed a Protein- and/or Phosphorus-reduced Diet” is significant in this field of interest. The manuscript is well-structured with enough data. However, this manuscript has minor issues needed to be addressed. Thus I recommend this manuscript for minor revision. 

  1. All table titles are too long. Please shorten the title to fit into a single line and provide the remaining information as a table footnote/legend. For example: DM (dry matter), N (nitrogen), and P (phosphorus). Mean values±standard errors; n = 7 animals. a,b,c Mean values within a row with different superscript letters were significantly different; Tukey’s multiple comparisons test (p < 0.05).

Author: We have shortened the titles and placed the relevant information in the footnotes. Unfortunately, due to the given format of the IJMS-template, it was not possible to limit the information to one line everywhere.

Line 109, 114, 135, 162, 171, 197, 217

  1. The legends for Figures 1 and 2 should be moved above the graphs.

Author: We moved the legends for Figure 1 and 2 above the graph.

Line 167, 177

  1. Tables 8, 9, 10, and 11 should be converted into supplementary tables.

Author: We have converted Table 8 into supplementary materials. The authors believe that Tables 9, 10, and 11 (new 8, 9, 10) should remain in the manuscript for direct reference to the methodology. Moreover, the IJMS requires that authors publish the full experimental details so that the results can be reproduced. These tables are located in the section “Material and Methods” at the end of the manuscript and thus do not interfere with the flow of reading.

  1. The manuscript must be subjected to a typographical error check. For example, "concentrate feed und wheat straw" should be presented as "concentrate feed and wheat straw."

Author: The manuscript was checked for typographical errors. Typographical errors could be eliminated in this way.

  1. I suggest conducting a PCA analysis of the data to make it easier to understand the changes in EAA (essential amino acids) and LEAA (non-essential amino acids).

Author: Your suggestion to do a principal component analysis (PCA) is very interesting as this is the best method to reduce the dimensionality of data. PCA is particularly useful when the number of variables is large, and it is difficult to interpret or visualize the data. However, the effects of protein reduction on essential and non-essential amino acids can be interpreted well in this study, and we believe that the heatmap best represents the different effects. In the last few days, we have been looking into the possibility of a PCA. However, a meaningful presentation of the data with this method was not possible due to time constraints. We will keep this method in mind for the following projects.

  1. Kindly shorten the conclusion

Author: The conclusion has been clarified and shortened.

We would like to thank you once again for your helpful comments and suggestions for improving the manuscript, and hope that the change was in your interest.

Kind regards,

Sarah Weber